# Anti-inflammatory activity of soluble chito-oligosaccharides (CHOS) on VitD3-induced human THP-1 monocytes

Paiboon Jitprasertwong[1☯], Munthipha Khamphio[2☯], Phornsiri Petsrichuang[2☯], Vincent G. H. Eijsink[3], Wanangkan Poolsri[4], Chatchai Muanprasat[4], Kuntalee Rangnoi[2], Montarop Yamabhai[2]*

1 School of Geriatric Oral Health, Institute of Dentistry, Suranaree University of Technology (SUT), Nakhon Ratchasima, Thailand, 2 Molecular Biotechnology Laboratory, School of Biotechnology, Institute of Agricultural Technology, Suranaree University of Technology (SUT), Nakhon Ratchasima, Thailand, 3 Faculty of Chemistry, Biotechnology and Food Science, Norwegian University of Life Sciences (NMBU), Ås, Norway, 4 Faculty of Medicine, Chakri Naruebodindra Medical Institute, Ramathibodi Hospital, Mahidol University, Samutprakarn, Thailand

☯ These authors contributed equally to this work.
* montarop@g.sut.ac.th

**Data Availability Statement:** All relevant data are within the manuscript and its Supporting information files.

## Abstract

Chito-oligosaccharides (CHOS) are oligomers of D-glucosamine and *N*-acetyl-glucosamine. Anti-inflammatory activities of a wide variety of CHOS mixtures have previously been reported, mainly based on studies with mouse models and murine macrophages. Since the mouse and human immune systems are quite different, gaining insight into the activity of CHOS on human cell lines, using well-characterized CHOS mixtures, is of considerable interest. *Bacillus subtilis* chitosanase (BsCsn46A) can efficiently convert chitosan to mixtures of water soluble low molecular weight CHOS. Here, the anti-inflammatory activity of a properly characterized CHOS mixture was studied, using human THP-1 cells that were differentiated to mature monocytes using vitamin D3. Addition of CHOS reduced the production of multiple pro-inflammatory cytokines associated with bacterial lipopolyssacharide (LPS)-stimulated inflammation, in a dose-dependent manner and without affecting cell viability. Interestingly, only minimal effects of CHOS were observed in similar experiments with phorbol 12-myristate 13-acetate- (PMA-) differentiated, macrophage-like, THP-1 cells. Altogether, in addition to showing promising biological effects of well-characterized low molecular weight soluble CHOS in a human system, the present study also points at Vitamin D3-stimulated THP-1 cells as a favorable system for assessing the anti-inflammatory activity of bioactive compounds.

## Introduction

Chitosan, a deacetylated form of chitin, which is the main structural component of the exo-skeletons of arthropods and fungi, can be considered as one of the most promising biomaterials of the 21st century [1]. This is because chitosan and its derivatives possess diverse biological

**Funding:** This research was supported by the Suranaree University of Technology (SUT) [grant no. RU12/2562], the National Research Council of Thailand (NRCT) and the Office of the Higher Education Commission (OHEC) under the National Research University (NRU) project [grant no. SUT3-304-55-36-18 and SUT3-304-62-36-18]. MK and KR were supported by SUT Fulltime Doctoral Research [grants no. 61/13/2561 and 61/28/2563] and Thailand Science Research and Innovation (TSRI). PP was supported by a SUT-PhD scholarship.

**Competing interests:** The authors have declared that no competing interests exist.

activities, including antioxidant, anti-inflammatory, anti-diabetes, and anti-bacterial activities [2]. Indeed, multiple companies are currently aiming at commercializing chitin-derived products for a wide variety of applications, ranging from environmental, agricultural, food, medical, pharmaceutical, and cosmeceutical applications [3]. Nevertheless, the mechanisms of action of different chitosans and chitosan-derivatives remain often unclear, one reason being that the compositions of the products and the biological assays used to test them vary [4].

Among the chitin-derived products, chitosan oligosaccharides (CHOS or COS) are highly attractive due to multiple favorable properties. For example, CHOS are soluble in mild acid, non-toxic, biocompatible, and biodegradable, and show good absorption to intestinal epithelia. CHOS can be generated by degradation of chitosan using different methods including heating in mild acid, microwave treatment, or enzymatic degradation [5]. The biological activity of CHOS has been shown to be closely associated with oligomer properties, which are defined by the degree of polymerization (DP), molecular weight (MW), the degree of $N$-acetylation ($D_A$), or the fraction of acetylation ($F_A$), and the pattern of $N$-acetylation ($P_A$) [6,7]. Enzymatic degradation of chitosan is an environmentally friendly approach to produce CHOS with well-defined properties. Various chitosan degrading enzymes have been characterized and can be used to produce CHOS by carefully controlling the biocatalytic process [8].

One of the most significant biological activities of CHOS is their immune-modulatory activity. So far, most studies on the immunomodulating activities of chitosan and CHOS were done using mouse models [9,10] or murine macrophages [10–12]. Until now, data from studies on human systems are scarce, but these limited data have shown interesting results. For example, a hepatoprotective effect of CHOS on Chang liver cells has been reported [13]. In addition, relative long CHOS (5–14 kDa) have been shown to inhibit nuclear factor kappa-light-chain-enhancer of activated B cells (NF-κB) transcriptional activity, NF-κB-mediated inflammatory responses, and barrier disruption via 5' adenosine monophosphate-activated protein kinase (AMPK)-independent mechanisms in the human colon [14]. Since mouse and human immune systems are quite different and considering the encouraging results of previous studies on human cells, gaining further insight into the activity of CHOS on human immune cells is of interest.

The human monocytic cell line THP-1 is the most widely used model for primary human monocytes and macrophages [15]. In this study, THP-1 cells were differentiated to both mature monocytes and microphage-like cells after which the effect of CHOS on Lipopolysaccharides (LPS)-induced proinflammatory responses was investigated. To do so, a well-defined mixture of low molecular weight (0.4–1.4 kDa) soluble CHOS was generated through enzymatic hydrolysis of a highly deacetylated chitosan ($F_A$ 0.15) with a previously characterized *Bacillus subtilis* chitosanase (*Bs*CsnA) [16].

## Materials and methods

### Chitosans

Chitosan $F_A$ 0.15, with a MW of approximately 37 kDa, was provided by Teta Vennrensing (Kløfta, Norway).

### Chitooligosaccharide (CHOS) production and analysis

Production of CHOS was performed based on a previous method [17]. Briefly, one gram of chitosan was dissolved in 1% HCl containing 0.1 mg/mL bovine serum albumin (BSA). Then, the pH was adjusted to 5.5 using 1M NaOH. Chitosan solutions (final concentration of 10 mg/mL, in 100 mL reaction volume) were pre-incubated for 30 min in an incubator shaker at 37˚C. The degradation reaction was started by adding 0.5 μg of *Bacillus subtilis* chitosanase

(*Bs*Csn46A; [16]) per mg of chitosan. To obtain maximum degradation, the same amount of enzyme was added at 6 h (360 min) and 24 h (1,440 min) after the start of the incubation, which was continued for 48 h (2,880 min) in total. Samples were taken from the reaction at various time points, from 10 min to 2,880 min. The reaction was stopped by adding 1M HCl followed by heating at 100°C for 3 min. Then, the samples were dried under vacuum. For characterization, the samples were dissolved in $D_2O$ or 0.15 M ammonium acetate, pH 4.5, for $^1$H-NMR or size exclusion chromatography (SEC) analysis, respectively. The remaining reaction was stopped by heating in boiling water for 10 min to inactivate the enzyme. Then, the CHOS were lyophilized and stored at -20°C until use. For biological assays, CHOS were dissolved in de-ionized water (DI) and the pH was adjusted with 6 N NaOH to 7.

## NMR spectroscopy

Samples for $^1$H-NMR analysis were dissolved in $D_2O$, and the pD was adjusted to 4.3–4.6 with DCl or NaOD. $^1$H-NMR spectra were obtained at 400 M*Hz*, at 85°C, as previously described [18]. The degree of scission ($\alpha$) and the DPn were calculated according to a previously published method [19].

## Size exclusion chromatography (SEC)

CHOS were separated on three XK 26 columns connected in series and packed with Superdex™ 30 (Pharmacia Biotech, Uppsala, Sweden), with an overall dimension of 2.60 × 180 cm. The elution buffer used was 0.15 M ammonium acetate, pH 4.5. The elution buffer was pumped through the system using an LC-10ADvp pump (Shimadzu GmbH, Duisburg, Germany), delivering the elution buffer at a flow rate of 0.8 mL/min. The eluting oligomers were monitored with a refractive index (RI) detector (Shodex RI-101, Shodex Denko GmbH, Dusseldorf, Germany) coupled to a CR 510 Basic Data logger (Campbell Scientific Inc., Logan, UT). To characterize the isolated CHOS, fractions of 4 mL were collected using a fraction collector. For quantitative studies of degradation, typically 10 mg of degraded chitosan was injected.

## Analysis of CHOS by mass spectrometry (MS)

Identification of CHOS was performed with Matrix-Assisted Laser Desorption Ionization mass spectrometry (MALDI TOF/TOF MS). MS spectra were acquired using an UltraflexTM TOF/TOF mass spectrometer (Bruker Daltonik GmbH, Bremen, Germany) with gridless ion optics under the control of Flexcontrol. For sample preparation, 1 μL of the reaction products was mixed with 1 μL of 10% 2,5 dihydroxybenzoic acid (DHB) in 30% acetonitrile and spotted onto a MALDI target plate (Bahrke et al. 2002). The MS experiments were conducted using an accelerating potential of 20 kV in the reflectron mode.

## Cell culture of THP-1 monocytes

THP-1 cells (passage 41 Lot-no. 300356-1714SF) were purchased from CLS Cell Lines Service GmbH, Eppelheim, Germany. The cells were maintained in Roswell Park Memorial Institute (RPMI) 1640 medium, supplemented with 1% L-glutamine, 1% sodium pyruvate, and 10% fetal bovine serum (Gibco, Barcelona, Spain) at a concentration of 3–8 x $10^5$ cells/mL, at 37°C, 5% $CO_2$. The medium was changed three times a week and cells were passaged at regular intervals to assure optimum cell concentration. Cells were counted under a microscope using a haemocytometer (Bright-Line, improved Neubauer, Hausser Scientific, Pennsylvania, USA) and passaged at least one time after thawing before being used for an experiment. All experiments were conducted with cells after 4–14 passages.

## Differentiation of THP-1 cells with Vitamin $D_3$ or Phorbol 12-myristate 13-acetate (PMA)

THP-1 monocytes ($1 \times 10^6$ cells/mL) in growth media were pre-mixed with 0.2 μM 1α, 25-Dihydroxy-Vitamin $D_3$ (VitD3, Calbiochem, Merck Chemicals, Nottingham, UK) or 25 ng/mL (40 nM) PMA (Cat. P1585, Sigma-Aldrich, Missouri, USA) for inducing differentiation into mature monocytes or macrophages, respectively, before seeding into appropriate tissue culture plates. Unless otherwise stated, 400 μl of the cell mixture was seeded into wells of a 24-well tissue culture plate ($4 \times 10^5$ cell/well). After 48 h, VitD3 differentiated THP-1 cells were used for a challenging experiment. For PMA-induced THP-1 cells, after 48 h, the plate was centrifuged at 700 $x$ $g$ for 5 min, after which the PMA-containing medium was replaced with fresh RPMI, and the plate was incubated for another 24 h at 37°C, 5% $CO_2$ before a challenging experiment was initiated.

## Treatment of CHOS before challenging differentiated-THP-1 cells with bacterial lipopolysaccharides (LPS)

To examine the effect of CHOS on the release of cytokine(s) by THP-1 cells upon LPS stimulation, THP-1 cells ($4 \times 10^5$ cell/well in 24-well plate) were differentiated with VitD3 or PMA as described in section 2.7. Then, 50 μL of a CHOS mixture in RPMI was added to reach final concentrations of 0 to 200 μg/mL, and the cells were incubated for 24 h. After that 50 μL of LPS from *E. coli* 0111:B4 (Invivogen, Calne, Wilts, UK) in RPMI medium to a final concentration of 100 ng/mL was added to stimulate the inflammatory response. After 7 h of LPS challenge, supernatants were collected to determine cytokine release as described in section 2.9. Dexamethasone (Cat. D4902, Sigma-Aldrich, Missouri, USA) at 0.2 μg/mL (0.5 μM), a standard anti-inflammatory drug, was used as a positive control in these experiments.

In one experiment, CHOS was removed from the cells after 24 h treatment before LPS was added. In this experiment, 500 μl of PBS was added, the plate was centrifuged at 700 $x$ $g$, supernatant was removed, and the cells were washed one more time with 1000 μl of PBS, before 500 μL of medium containing LPS at a final concentration of 100 ng/mL was added.

## Determination of cytokines released upon LPS-stimulation

ELISA-based methods were used to determine the concentration of cytokines secreted into culture supernatant after LPS stimulations of differentiated-THP-1 cells at various experimental conditions. The analysis was done according to the manufacturer's protocols (R & D Systems; Minneapolis, USA), for Interleukin (IL)-1β (Human IL-1β DuoSet ELISA, Cat.DY201, Lot.P218283), IL-6 (Human IL-6 DuoSet ELISA, Cat.DY206, Lot.P154822) and Tumor necrosis factor (TNF)-α (Human TNF-α DuoSet ELISA, Cat.DY210, Lot.P153726). The reported values represent the average of triplicate wells.

## The effect of CHOS on the viability of LPS-treated THP-1 monocytes (MTT assay)

THP-1 cells in 100 μL RPMI medium ($1 \times 10^5$ cells/well in a 96-well plate; triplicates) were treated with 0.2 μM VitD3 for 48 h. After that, 12.5 μL of CHOS solutions in RPMI medium to various final concentrations were added and incubated for 24 h. Then, 12.5 μL of LPS solution in RPMI medium to a final concentration of 100 ng/mL was added into each well. After 7-h of LPS stimulation, the plates were centrifuged at 700 x $g$ for 5 min, and 100 μL of the supernatant was removed. Then, the 3-(4,5-Dimethylthiazol-2-yl)-2,5-Diphenyltetrazolium Bromide (MTT) assay [20] was performed by adding 100 μL of a 0.5 mg/mL MTT (Invitrogen, USA)

solution in RPMI medium into each well. After 2 h of incubation at 37°C and 5% $CO_2$, the plate was centrifuged at 700 x $g$ for 5 min and the supernatant was removed. Then, formazan crystals were solubilized by adding 175 μL of solubilization reagent (40% Dimethylformamide, 20% glacial acetic 16% SDS, pH 4.7), followed by incubation at 37°C for 1 h. The absorbance at 595 nm was measured using a microplate reader (Tecan, Austria). The reported values are the average absorbance of three independent experiments after subtraction of a blank without cells.

## Assessment of cluster of differentiation (CD) 14 and cell viability by flow cytometry

THP-1 monocytes (2 x $10^6$ cell/well in 6-well plates) were differentiated with VitD3 or PMA, as described in section 2.7. To prepare cell for flow cytometry analysis, both suspension and attached cells were collected from the plate at 0, 48, 72 and 96 h into centrifuge tubes and spun down. Then 100 μL of cell suspensions in microcentrifuge tubes containing $2x10^5$ cells were incubated with 2 μL of FITC-conjugated anti-human CD14 Antibody (BioLegend Way, California, USA) for 15 min at 4°C, followed by washing with 900 μL of PBS. Then, the tubes were centrifuged at 700 x $g$ for 5 min to remove the supernatant, and, subsequently, the cells were stained with 1 μL of Propidium iodide (PI) (Cat. P3566, Life Technologies Corporation, USA) in 400 μL of PBS. For flow cytometry, a minimum of 10,000 events were acquired on an Attune™ NxT Acoustic Focusing cytometer (Thermofisher Scientific, USA), using Attune™ NxT v3.1.2 Software for data analysis. CD14 and PI were analyzed using the BL-1 and YL-2 channels, respectively. Quadrant regions indicated the percentage of cells for each sub-population.

## Effect of CHOS on AMP-activated protein kinase (AMPK) activation in T84 cells

Human colon T84 cells (American Type Culture Collection, Manassas, VA, USA) were cultured in a mixture of Dulbecco's Modified Eagle Medium (DMEM) and Ham's F-12 medium (1:1) (Gibco, MA, USA) supplemented with 10% fetal bovine serum (FBS), 100 U/mL penicillin and 100 μg/mL streptomycin (Gibco, Massachusetts, USA) in a 5% $CO_2$ humidified incubator at 37°C. For this assay, the cells were seeded in 6-well plates (Corning, Manassas, Virginia, USA) at a density of $1 \times 10^6$ cells/well. After treatments with solubilizing reagent (DMSO) or 100 μg/mL CHOS, or 10 μM A-769662 (a known AMPK activator; Tocris Bioscience, Bristol, UK) for 24 h, cell lysates were generated using a lysis buffer containing 50 mM Tris-HCl pH 7.4, 150 mM NaCl, 1 mM EDTA, 1% Triton-X100, 1 mM NaF, 1 mM $Na_3VO_4$ and 1 mM PMSF. The protein concentration in the lysates was quantified using the Bradford assay (Bio-Rad Laboratories, California, USA). Absorbance was determined at 595 nm using a microplate reader (BioTek Synergy Neo2). Extracted proteins were separated by 10% sodium dodecyl sulphate polyacrylamide gel electrophoresis (SDS-PAGE) and transferred to 0.45 μm nitrocellulose membranes. The membranes were blocked with 8% non-fat milk (Bio-Rad Laboratories, California, USA) in Tris Buffered Saline and 0.1% Tween 20 (TBST) for one hour at room temperature, followed by overnight incubation at 4°C with specific primary antibodies against phospho-AMPKα (Thr172), AMPKα or β actin (Cell Signaling Technology, USA). Then, the membranes were incubated with a horseradish peroxidase (HRP)-conjugated goat anti-rabbit IgG secondary antibody (Abcam, Massachusetts, USA) for one hour at room temperature, followed by addition of the HRP substrate for enhanced chemiluminescence (Luminata Forte, Merck Millipore, Germany). The intensity of the protein bands was measured using a

ChemiDoc Imaging System (Bio-Rad Laboratories, California, USA). A total of 5 experiments were performed.

## Statistical analysis

The data was recorded as mean ± standard deviation (SD) and were analyzed by one-way and two-way analysis of variance (ANOVA). Kruskal-Wallis test and Dunn's multiple comparisons test were applied for analyses of non-parametric data of Figs 4–7. For Fig 3, parametric data were analyzed with Brown-Forsythe ANOVA and Unpaired t with Welch's correction test, after normal distribution was indicated. Statistical testing was performed using GraphPad Prism 8 software (GraphPad Software Inc., California, USA). Tables of data with mean ± standard deviation (SD), sample sizes and the exact *P*-values of each experiment can be found in the supplementary material, S1 Table.

# Results and discussion

## Characterization of the soluble CHOS

*Bs*Csn46A is one of the fastest chitosanases published so far, with reported initial rates being as high as $5.5 \times 10^3$ $min^{-1}$ for $F_A$ 0.15 chitosan [17]. A detailed analysis of CHOS generated by treating $F_A$ 0.15 chitosan with this enzyme has been reported previously [17]. In this study, the reaction was carried out in 100 mL scale and with multiple additions of higher amounts of enzyme to ensure that the chitosan substrate was completely converted to soluble CHOS.

To compare with the previous results obtained from the production of CHOS in analytical scale, $^1$H-NMR spectroscopy was used to analyze the time course of the degradation. Fig 1 shows the degree of scission (α), i.e., the fraction of glycosidic linkages in the chitosan that has been cleaved by the enzyme, plotted against the reaction time. The progress curve was similar to the curve obtained previously [17] and comprised 2 phases, a rapid initial phase, followed by a slower second phase. After 48 h, the α value reached the maximum of 0.37, which is slightly higher than the value typically obtained in analytical scale reactions. Size exclusion chromatography (SEC) was also employed to analyze the CHOS mixtures as show in Fig 2. At 24 h (1,440 min) of reaction, a small void peak (DP > 50) was still visible (Fig 2A). After addition of more enzyme and incubation for another 24 h, the polymer peak disappeared, confirming complete degradation of the chitosan substrate. The major products were DP 2–3 (Fig 2B) as previously observed [17].

Finally, the chemical compositions of CHOS in fractions obtained after the SEC were analyzed by MALDI-TOF-MS, and the results are summarized in Table 1. The data show that the hydrolytic products obtained upon hydrolyzing $F_A$ 0.15 chitosan with *Bs*Csn46A mainly are fully deacetylated dimers and trimers, whereas the less abundant longer products contain one or more acetylated units. The MS analysis confirmed the chromatographic data (Fig 2) showing that the CHOS products are small with the MW ranging from approximately 0.4 kDa (DP2) to 1.2 kDa (DP6).

## Different effects of CHOS on LPS-induced secretion of IL-1β by PMA or VitD3-differentiated THP-1 cells

Human THP-1 cells, induced for differentiation by PMA or VitD3 to become macrophages or mature monocytes, respectively, were used to explore a possible inhibitory effect of CHOS on LPS-stimulated release of the pro-inflammatory cytokine IL-1β. After differentiation, the cells were treated with 100 μg/mL of CHOS for 24 h before challenging with LPS for 7 h. Supernatants were collected and assayed for IL-1β concentrations by ELISA (Fig 3). The results showed

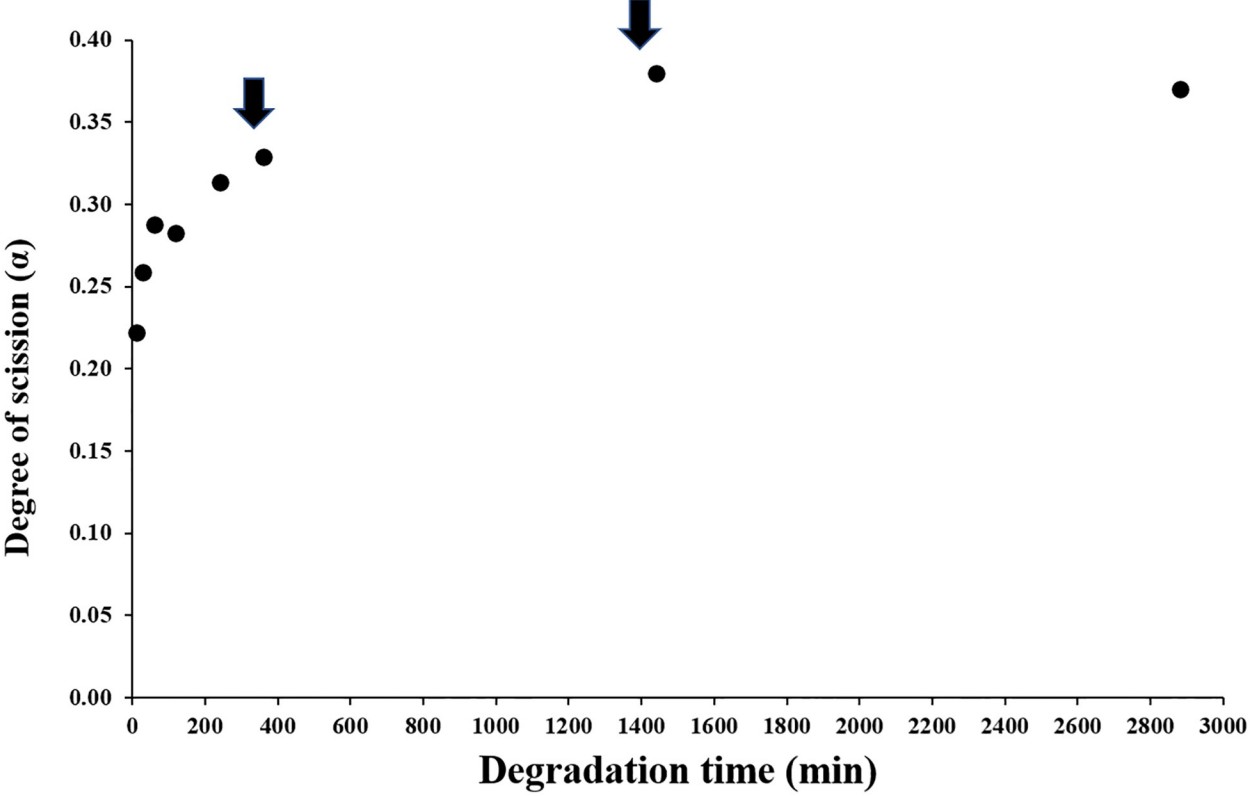

**Fig 1. Time-course of the degradation of chitosan by *Bs*Csn46A.** Time-course for the increase in the degree of scission ($\alpha$) during degradation of chitosan with $F_A$ 0.15 using *Bs*Csn46A. The graph shows the degree of scission ($\alpha$) determined by $^1$H NMR in 30 mL reactions containing 10 mg/mL chitosan and 0.5 µg of *Bs*Csn46A per mg of chitosan in 40 mM sodium acetate buffer, pH 5.5, 100mM NaCl, and 0.1 mg BSA/mL. The $\alpha$ value was calculated as the inverse value of $DP_n$ (average degree of polymerization) [19]. In an attempt to reach maximum conversion of the chitosan, additional *Bs*Csn46A was added at 0.5 µg per mg of chitosan after incubation for 360 min and 1440 min (black arrows), and the degradation reaction was continued for another 1440 min.

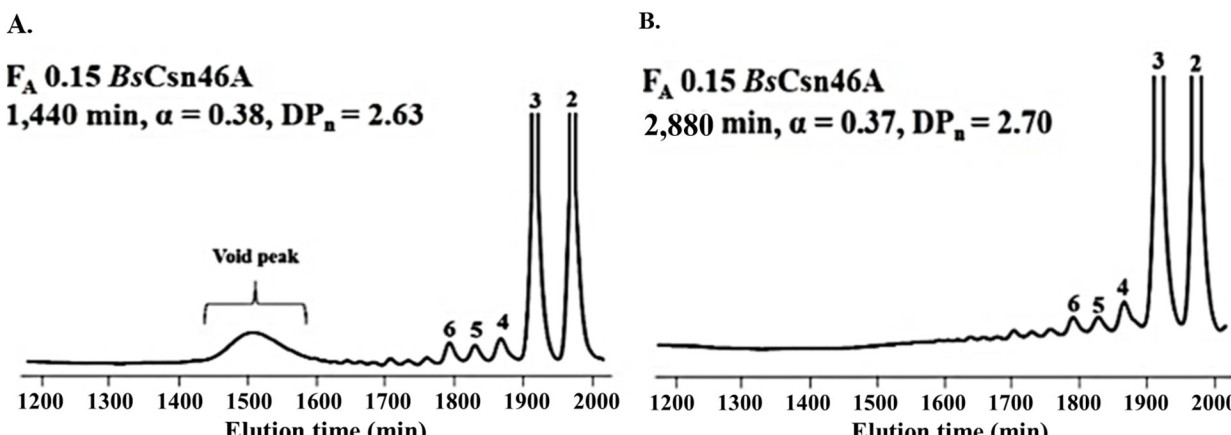

**Fig 2. Product analysis by size exclusion chromatography (SEC).** Size distribution of CHOS after degradation of chitosan $F_A$ 0.15 with *Bs*Csn46A at 37˚C for 24 h (A) and 48 h (B). The peaks are labeled by numbers indicating the DP values.

**Table 1. Chemical composition of CHOS fractions derived from the CHOS mixture used in this study[*].**

| DP | Detected species |
|---|---|
| 2 | **DD**, DA, AA |
| 3 | **D3**, **D2A1**, D1A2 |
| 4 | **D4, D3A1**, D2A2, |
| 5 | D5, **D4A1**, D3A2, D2A3 |
| 6 | D6, **D5A1**, **D4A2**, D3A3 |

[*]The CHOS preparation used in this study was fractionated by SEC (Fig 2B) and fractions containing oligomers with DP2 to DP6 were collected for analysis by MALDI-TOF MS. The bold letters represent seemingly dominant products in the fractions, based on the signal intensities in the MS spectra.

that, as expected, LPS induced IL-1β release from both PMA- and VitD3-differentiated THP-1 cells (grey bars) and that IL-1β levels were about two-fold higher for vitD3-differentiated THP-1 cells, compared to PMA-differentiated cells. Interestingly, treatment with CHOS inhibited IL-1β production. While the effect was modest for the PMA-induced cells, strikingly, the effect was large for the VitD3-induced THP-1 cells.

## Divergent responses of THP-1 cells upon induction for differentiation by PMA and VitD3

To gain insight into the discrepancy in the inhibitory effects of CHOS on LPS-induced IL-1β secretion by PMA- and VitD3-induced THP-1 cells, the expression of CD14, a macrophage biomarker, and cell viability, as a proxy of apoptotic cell death, were assessed. CD14 is an

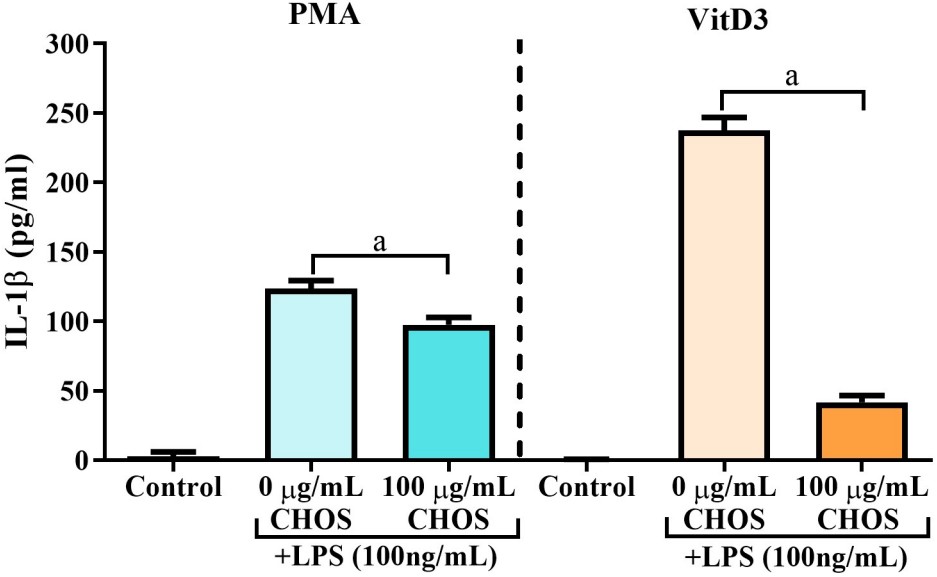

**Fig 3. The effect of CHOS on IL-1β release from PMA and VitD3-differentiated THP-1 monocytes/macrophages after challenging with LPS.** ELISA results showing IL-1β released after LPS stimulation of PMA- or VitD3-differentiated THP-1 cells. IL-1β was measured for cells that were treated with 0 and 100 μg/mL CHOS for 24 h before challenging with 100 ng/mL LPS for 7 h. The IL-1β concentration for cells that were not challenged with LPS (control) is shown for comparison. The results are expressed as the mean ± SD of duplicates from three independent experiments (n = 6); a indicates significant difference at ($P < 0.001$).

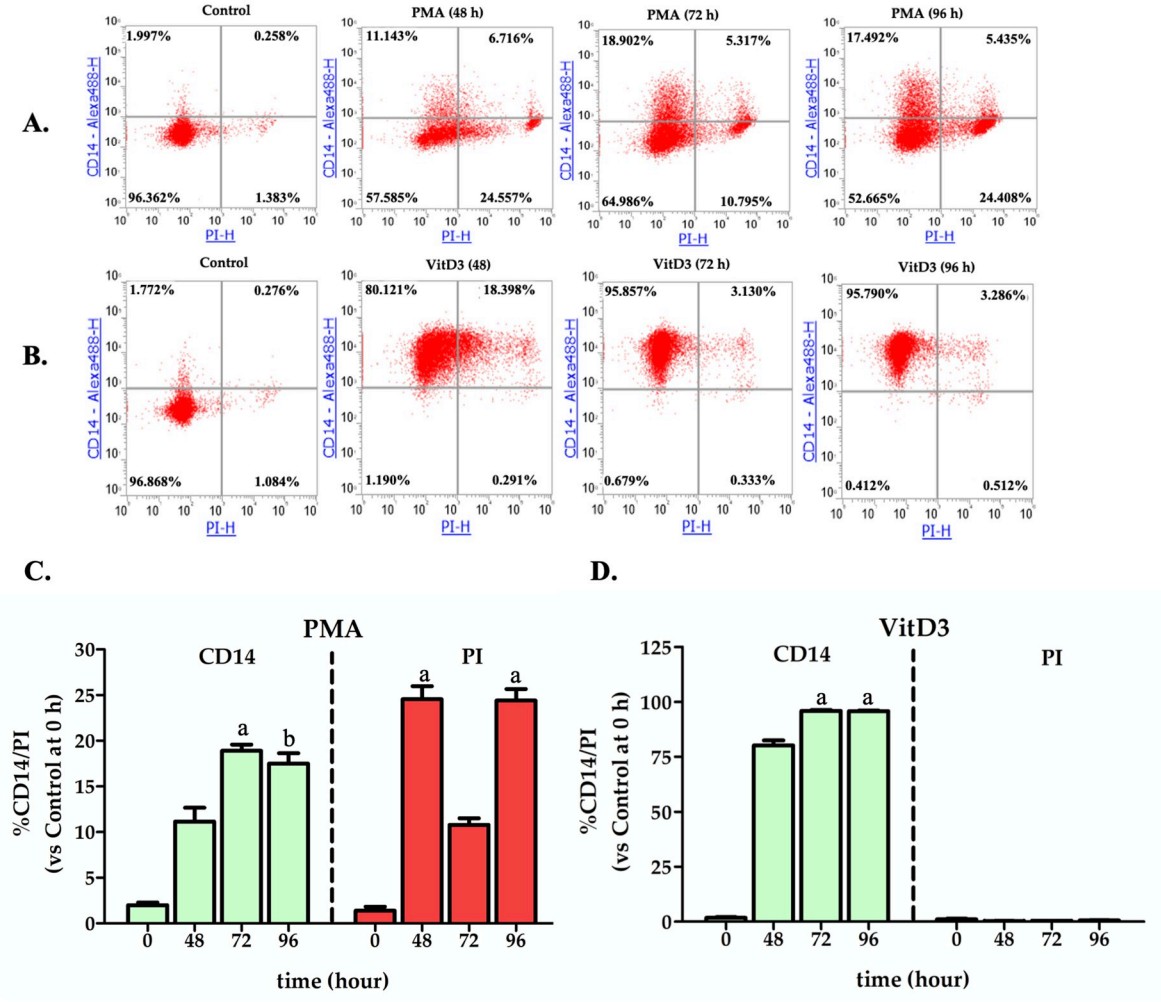

**Fig 4. Flow cytometry analysis of CD14 expression and PI staining.** Panels (**A**) and (**B**) show representative experiments of flow cytometry dot plots for PMA- and VitD3- differentiated cells, respectively. The bar graphs show analyses of CD14 expression and PI staining for PMA- (**C**) and VitD3-differentiated (**D**) THP-1 cells, measured by flow-cytometry at different time points, namely 0, 48, 72 and 96 h. The percentages of CD14- and PI- stained cells in comparison to the un-stimulated THP-1 cells (time 0) are shown. The bars show the mean ± SD of three independent duplicated experiments (n = 6); a = ($P < 0.001$) and b = ($P < 0.01$) indicate significant differences, relative to the control (0 time). Note that for PMA treated cells, the medium containing PMA was removed after 72 h to fresh media in order to get rid of PMA.

essential co-receptor for the recognition of bacterial LPS, binding directly to LPS and presenting LPS molecules to the TLR4-MD-2 signaling complex, which leads to cellular activation [21]. These two properties were assessed by antibody and propidium iodide (PI) staining, respectively, and staining levels were determined by flow cytometry. THP-1 monocytes (1.0 x $10^6$ cells/well) were incubated with PMA or VitD3 and collected for staining and subsequent analysis at 4 different time points, i.e., at 0, 48, 72 and 96 h after induction. As shown in Fig 4, a large increase in CD14 expression was observed only in vitD3-induced THP-1 monocytes. Notably, while PMA had lesser effect on CD14 expression, cell death amounting to 10–25% of the cells was observed, as indicated by PI staining (Fig 4A).

In summary, Fig 4 shows that the different responses to LPS and CHOS in VitD3- and PMA-induced cells may be due to quite pronounced differences between the two differentiated cell types. It is known that stimulation of THP-1 cells with potent inducer agents such as PMA

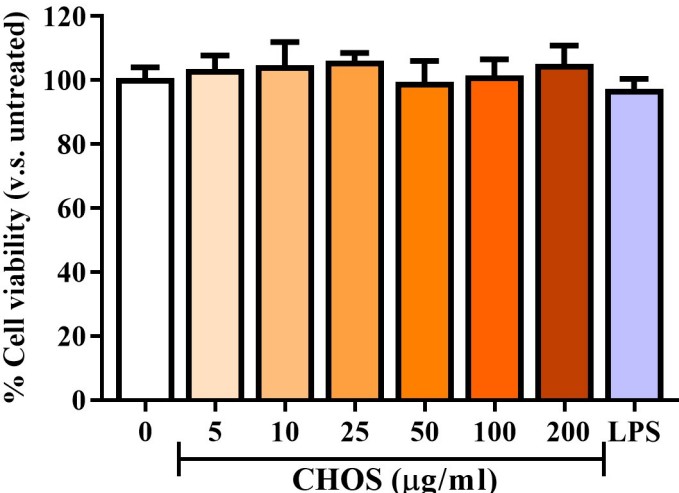

**Fig 5. Cytotoxic effects of CHOS and LPS on VitD3-differentiated THP-1 cells.** The MTT assay was used to determine cell viability after VitD3-differentiated THP-1 cells had been treated with various concentrations of CHOS for 24 h. Relative cell viability was calculated based on untreated cells (0 μg/ml, 100%). The effect of LPS on the viability of VitD3-differentiated THP-1 cells was also determined, as shown by the grey bar. The data shown are mean±SD from three independent triplicated experiments (n = 9). A Kruskal-Wallis test indicated no significant differences relative to cells treated with 0 μg/mL CHOS.

may cause phenotypic and functional differences and growth cycle arrest [22]. Accordingly, in the present experiments, the number of PI-positive cells increased after PMA- induced differentiation, suggesting that PMA may be cytotoxic, while there were almost zero PI-positive cells among vitamin D3-differentiated cells. Treatment of THP-1 cells with VitD$_3$ is known to induce the cells to differentiate along the myeloid lineage to mature monocyte-like cells, resembling primary human monocytes, which are characterized by expressing a high level of CD14 and are capable of releasing a myriad of cytokines [23,24].

Taken together, the present results demonstrate for the first time that vitamin D3-differentiated THP-1 cells provide an appropriate model system that can be used as an *in vitro* platform for screening of anti-inflammatory activities of bioactive compounds, including CHOS. Therefore, VitD3-induced THP-1 cells were used in the subsequent experiments.

## CHOS and LPS have no cytotoxic effects on THP-1 monocytes

The MTT assay was performed to evaluate the effects of CHOS and LPS on the viability of VitD3-induced THP-1 cells. The experiments were designed to resemble the standard anti-inflammatory assay. Cells were exposed to various concentration of CHOS (0–200 μg/mL) or LPS (100 ng/mL). As shown in Fig 5, neither CHOS (0–200 μg/mL) nor LPS affected the viability of the THP-1 monocytes. These results indicate that CHOS and LPS, at the cocenetrations used in this study have no direct effect on THP-1 monocytes, and that CHOS-induced changes in cytokine or cell surface receptor expression were not due to changes in cell viability or proliferation.

## Dose-response relationships for the inhibitory activity of CHOS on secretion of IL-1β, IL-6 and TNF-α by LPS-stimulated VitD3-differentiated THP-1 cells

To further evaluate the inhibitory effect of CHOS on the production of inflammatory cytokines by THP-1 monocytes, dose-response effects for CHOS-mediated inhibition of the

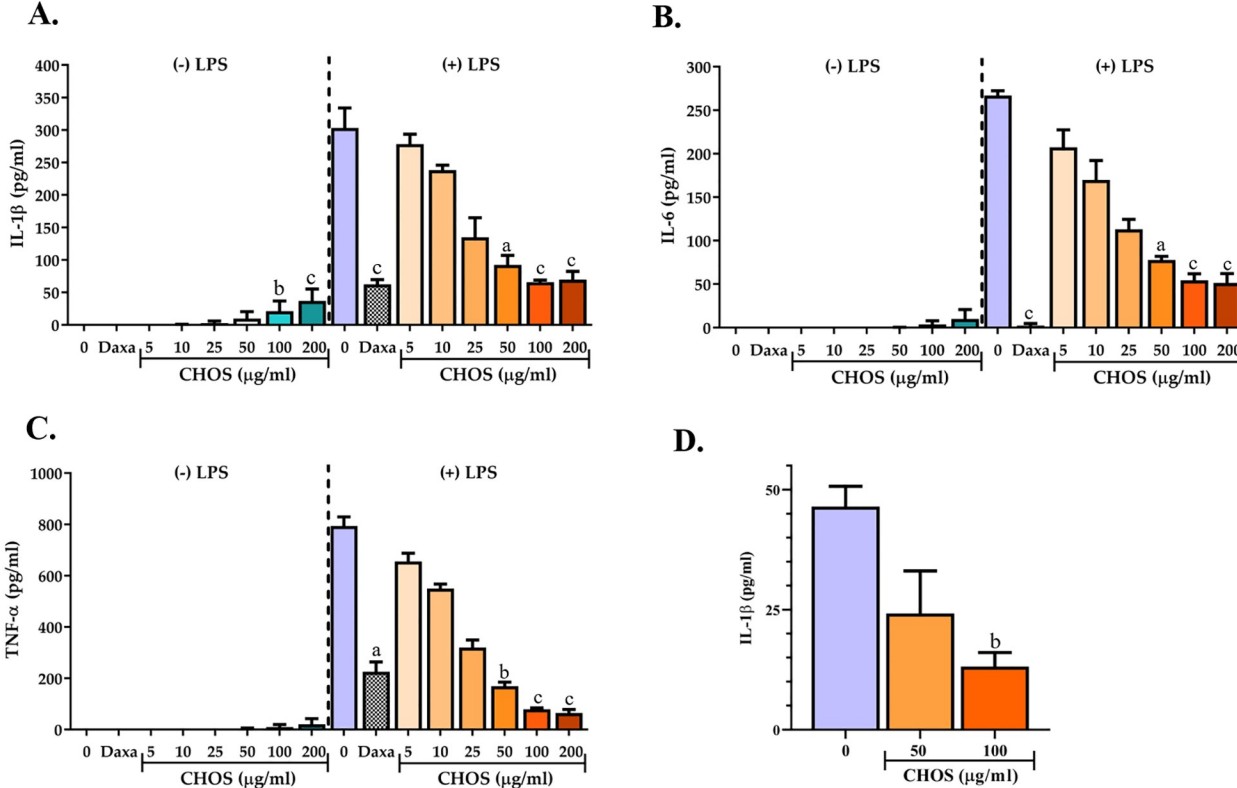

**Fig 6. Dose-repose effects for the *in vitro* anti-inflammatory activity of soluble CHOS on human THP-1-derived monocytes.** The effects of CHOS, at final concentrations ranging from 0 to 200 μg/mL, on release of IL-1β (**A**), IL-6 (**B**), and TNF-α (**C**) by LPS-stimulated, VitD3-differentiated THP-1 cells were determined by ELISA. Dexamethasone (0.5 μM), a standard anti-inflammatory drug, was used as a positive control. (**D**) An extra washing step was done after CHOS treatment before LPS challenging. The shown data are from three independent duplicated experiments (n = 6) except Fig 6D, which reflects one triplicated experiment (n = 3). Significance levels: a = *P* < 0.001, b = *P* < 0.01, c = *P* < 0.05 (compared with 0 μg/mL CHOS). See S1 Fig for an enlargement of data from the experiments without LPS stimulation (-LPS).

production of three pro-inflammatory cytokines, IL-1β, IL-6 and TNF-α, that are mainly produced by monocytes and macrophages [25], were determined. These cytokines are elevated in the early phase of infection and inflammation, and their excessive or unbalanced production may enhance systemic inflammation and mediate severe inflammatory diseases, which are responsible for the initiation and progression of a series of pathologies [26,27], including SARS-CoV-2 infection, which may lead to a lethal cytokine storm [28].

In this experiment, VitD3-differentiated THP-1 cells were treated with CHOS (0–200 μg/mL) for 24 h and then exposed to LPS for 7h. Dexamethasone-treated cells served as a positive control. The levels of IL-1β, IL-6 and TNF-α in the supernatants were analyzed by ELISA. Fig 6A–6C shows that LPS significantly increased the secretion of IL-1β, IL-6 and TNF-α from THP-1 monocytes (*p* < 0.001). Most importantly, CHOS significantly decreased LPS-induced IL-1β, IL-6 and TNF-α release from THP-1 monocytes (*p* < 0.001) in a dose-dependent manner. Anti-inflammatory activity of CHOS could be observed when CHOS concentrations were as low as 5 μg/mL and the effect gradually increased upon increasing the dose of CHOS, with maximum inhibition being reached at a dosage of 100 μg/mL. Importantly, the levels of all three pro-inflammatory cytokines were reduced. The maximum inhibitory effect obtained with these soluble CHOS is comparable to the effect of 0.2 μg/mL (0.5 μM) dexamethasone.

Notably, while CHOS at 200 μg/mL strongly inhibited the cytokine response upon LPS stimulation, this high concentration of CHOS slightly enhanced the release of IL-1β, IL-6 and

TNF-α in cells that were not exposed to LPS (see S1 Fig for an enlargement of the Y-axis scale). These results indicate that the optimal dosage of CHOS must be carefully evaluated to obtain the highest benefit, as the higher dosage of CHOS seems to (slightly) promote inflammation.

High molecular weight (30–130 kDa) chitosan forms a stable water-soluble chitosan–LPS complex through electrostatic interactions. Interestingly, it has been shown that these complexes enhance the secretion of TNF-α from murine monocyte cell line RAW 364.7 and IL-8 production from HEK293 cells overexpressing TLR4/MD2 [29]. This result contradicts another finding, which indicated that chitosan with a molecular weight of 5–300 kDa could slightly reduce LPS-induced secretion of TNF-α and IL-1β by RAW 364.76 cells, in a dose-independent manner [30]. Compared to these previous studies, with somewhat conflicting results, the CHOS used in the present study are very small and have a much stronger inhibitory effect on LPS-induced secretion of pro-inflammatory cytokines. Considering the small size and the performance of the CHOS, it is unlikely that complexation with LPS, and, consequently, blocking the effect of LPS on the THP- cells, is the mechanism behind the observed anti-inflammatory effects. To proof this hypothesis, an additional experiment with an extra washing step, after the CHOS treatment and before challenging with LPS, was conducted. As shown in Fig 6D, inhibitory effects of soluble CHOS can still be observed, even though the amount of secreted IL-1β was reduced due to a reduction in the numbers resulting from the washing step. This finding indicates that CHOS acts on an intracellular inflammatory signaling pathway of the THP-1 monocytes.

Anti-inflammatory activities of different CHOS mixtures with various compositions obtained from diverse preparation methods have been reported using different assay systems, such as mouse models and various cell lines including murine RAW 264.7 macrophages, murine *microglial* cell line *BV-2*, and human endothelial T84 cells [2,31]. To the best of our knowledge, this is the first report on the anti-inflammatory activity of CHOS on human monocytes.

## Soluble CHOS do not inhibit CD14 expression in THP- cells but activate AMP activated protein kinase (AMPK) in T84 cells

In an attempt to find the mechanism of the anti-inflammatory effect of CHOS, the effects of CHOS on the expression of CD14 by THP-1 cells and VitD3-differentiated THP-1 cells (monocytes) were determined using flow cytometry. As shown in Fig 7, CHOS slightly enhanced CD14 expression in VitD3-differentiated THP-1 monocytes, but this effect was minimal compared to the effect of the VitD3 induction itself on CD14 levels. Moreover, when VitD3-differentiated THP-1 cells were challenged with LPS, the levels of CD14 were not affected by treatment with CHOS. These results indicate that the anti-inflammatory effects of CHOS is not due to a reduction of CD14 levels.

Longer CHOS with MW of $> 5$ kDa have been reported to activate AMP-activated protein kinase (AMPK) [14], and this effect could, at least in part, explain the previously observed anti-inflammatory effect of such CHOS in human intestinal epithelial (T84) cells [2,14]. Thus, this biological activity was investigated using the soluble CHOS employed in the present study (100 µg/mL; 24 h incubation). The results showed that, these short CHOS did activate AMPK in T84 cells, similar to the effect of a known AMPK activator, A-769662, that was used as a positive control (Fig 8). Indeed, it has been reported that AMPK activation could suppress proinflammatory response in mouse and human macrophage [32]. Therefore, one of the possible mechanism of actions of the soluble CHOS is via the activation of AMPK.

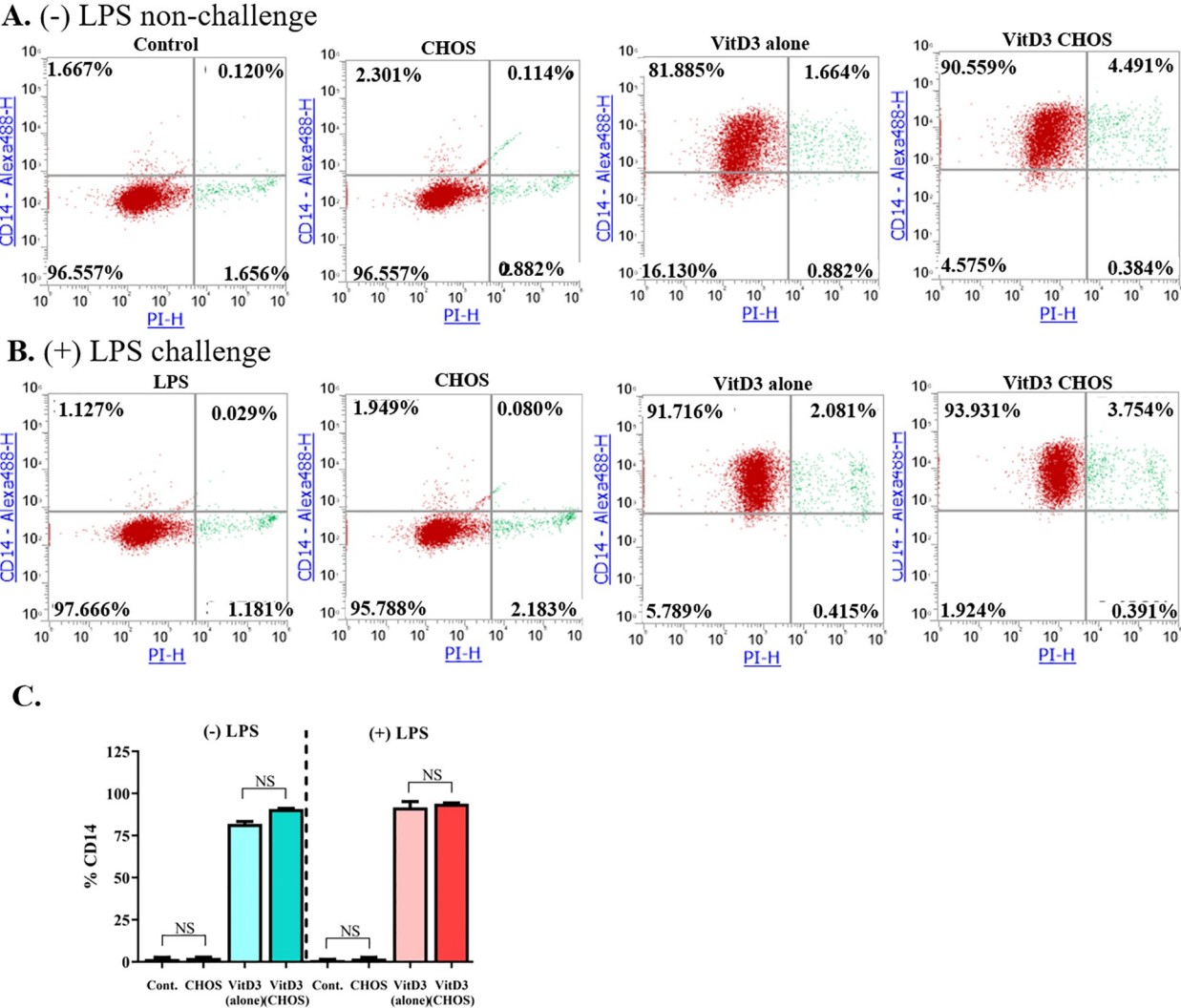

**Fig 7. The effect of CHOS and LPS on CD14 expression and viability in THP-1 cells.** Both non-differentiated and VitD3 differentiated THP-1 cells were treated with 100 μg/mL of CHOS and 100 ng/mL of LPS. Flow cytometry was used to determine CD14 expression and PI staining. Panels (**A**) and (**B**) show flow cytometry dot plots for representative experiments. The bar chart (**C**) is generated from the average of data from the upper-left quadrant of three independent experiments. The data represent three independent duplicated experiments (n = 6). NS means non-significant.

Inflammation and related cytokine production are complex processes and there are multiple inflammatory signaling pathways that can be inhibited by different compositions of CHOS mixtures. Further investigations must be conducted to completely understand the mechanism of the anti-inflammatory effects of the soluble CHOS used in the present study. Further analysis of the anti-inflammatory effects of CHOS in specific organs *in vivo* should be performed to further assess the potential of using CHOS for protective or therapeutic purposes.

## Study strength and limitations

The strength of the present study lies in the utilization of properly characterized CHOS that were prepared using (green) enzyme technology and that, in contrast to the majority of previous studies, were well characterized. The CHOS mixtures in this study have low MW, facilitating incorporation into various nutraceutical or cosmeceutical preparations. In addition,

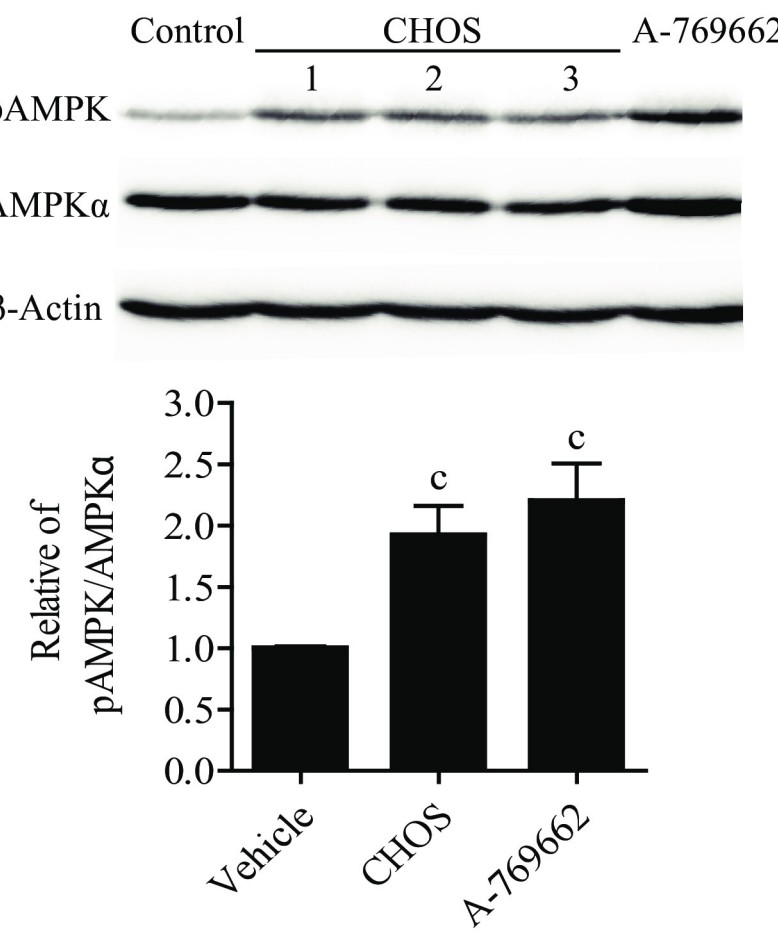

**Fig 8. Effects of CHOS on AMPK activation.** Western blot (WB) analysis of p-AMPK, AMPKα, and β-actin after treating T84 cells with CHOS at 100 μg/mL for 24 h. Data are analyzed as the ratio of p-AMPK/AMPKα and expressed as fold change in the ratio compared to the ratio for the control group (means ± SEM, n = 5). The compound A-769662 is a known activator of AMPK. c = P < 0.05 (compared with control).

human cells, as opposed to commonly used murine systems, were used for characterizing the CHOS, leading to the conclusion that soluble low MW CHOS only act on monocytes. The selective functionality of the CHOS confirms previous observations that different structures of CHOS can have different biological activities. While our study sheds light on the use of THP-1 cells for testing bioactivity, one limitation of the present study is that the biological effects of CHOS were only assessed *in vitro*. Moreover, transcriptomic analyses, which could help elucidate the mechanism of action of CHOS have not been done in this study. Such deeper analyses, including *in vivo* analysis in humans (clinical trials), are of great interest and needed to eventually bring CHOS to the clinic.

## Conclusion

In summary, the present study has demonstrated that enzymatic hydrolysis of highly de-acetylated chitosan ($F_A$ 0.15) by a chitosanase from *Bacillus subtilis* (BsCsn46A) yielded soluble, small CHOS products with potent anti-inflammatory activity when tested on VitD3-induced THP-1 cells. A similar effect of CHOS was not observed when using PMA-induced THP-1 cells. The results underscore that the choice of the differentiation method is important when

studying immunomodulatory activities of CHOS with human THP-1 monocytes and that different CHOS structures can have diverse biological activities. The strong anti-inflammatory activity of the CHOS studied here, comparable to Dexamethasone, implies clinical relevance as a novel non-steroidal anti-inflammatory substance. Importantly, the results also indicated that, while CHOS have promising medically relevant properties, the dosage of CHOS in potential pharmaceutical, nutraceutical or cosmeceutical products needs to be carefully optimized to obtain the highest benefits and prevent adverse reactions. Since chitin/chitosan and CHOS are biocompatible and have low toxicity, clinical trials of soluble CHOS products for prevention and treatment of pathogenic inflammatory conditions seem an attractive next step.

## Supporting information

**S1 Fig. Enlargement of scale for samples without LPS treatment (-LPS).** The results of Fig 6 are plotted with a different Y-axis scale, to highlight the values of samples without LPS treatment.
(TIF)

**S1 Table. Data from statistical analyses.**
(PDF)

**S1 File.**
(PDF)

## Acknowledgments

The authors would like to thank Chai Noi Soem, Thae Thae Min, Thanaporn Pimpakan, and Salinthip Jarusintanakorn for excellent technical assistance and advice.

## Author Contributions

**Conceptualization:** Paiboon Jitprasertwong, Vincent G. H. Eijsink, Montarop Yamabhai.

**Data curation:** Chatchai Muanprasat, Kuntalee Rangnoi.

**Funding acquisition:** Montarop Yamabhai.

**Investigation:** Paiboon Jitprasertwong, Munthipha Khamphio, Phornsiri Petsrichuang, Wanangkan Poolsri.

**Methodology:** Paiboon Jitprasertwong, Chatchai Muanprasat, Montarop Yamabhai.

**Project administration:** Montarop Yamabhai.

**Resources:** Montarop Yamabhai.

**Supervision:** Paiboon Jitprasertwong, Vincent G. H. Eijsink, Chatchai Muanprasat, Montarop Yamabhai.

**Validation:** Paiboon Jitprasertwong, Vincent G. H. Eijsink, Kuntalee Rangnoi, Montarop Yamabhai.

**Visualization:** Munthipha Khamphio, Phornsiri Petsrichuang, Wanangkan Poolsri.

**Writing – original draft:** Paiboon Jitprasertwong, Munthipha Khamphio, Phornsiri Petsrichuang, Chatchai Muanprasat.

**Writing – review & editing:** Vincent G. H. Eijsink, Montarop Yamabhai.

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
