## [Decision Letter · Decision Letter 0]

24 Sep 2020

PONE-D-20-27129

Anti-inflammatory activity of soluble chito-oligosaccharides (CHOS) on VitD3-induced human THP-1 monocytes

PLOS ONE

Dear Dr. Yamabhai,

Thank you for submitting your manuscript to PLOS ONE. After careful consideration, we feel that it has merit but does not fully meet PLOS ONE’s publication criteria as it currently stands. Therefore, we invite you to submit a revised version of the manuscript that addresses the points raised during the review process.

 As appended below, the reviewers have raised major concern/critique and suggested further experimental work to consolidate the findings. Do go through the comments and amend the MS accordingly.

We look forward to receiving your revised manuscript.

Kind regards,

A. M. Abd El-Aty

Academic Editor

PLOS ONE

Additional Editor Comments:

1- What are the final conclusion and take home message? Add this at the end of the abstract

2-  What are the study strengths and limitations? Add this in a separate section ahead of conclusion. Start with the strength of the study followed by limitation.  

3- Provide Figs in color 

4- What are the clinical relevance and future perspective? Add this to the conclusion section. The clinical relevance cannot be summarized in "more prospective studies are needed...", that has no clinical relevance. Please point at how your findings impact the care of patients. 

5- Make sure that Tables/Figs/references were cited in order without missing any 

6- Make sure that Table legends and Fig. captions are provided, without missing any

Journal Requirements:

Reviewers' comments:

Reviewer's Responses to Questions

**Comments to the Author**

1. Is the manuscript technically sound, and do the data support the conclusions?

Reviewer #1: Yes

Reviewer #2: Yes

Reviewer #3: Partly

Reviewer #4: Yes

Reviewer #5: Partly

2. Has the statistical analysis been performed appropriately and rigorously? 

Reviewer #1: Yes

Reviewer #2: Yes

Reviewer #3: Yes

Reviewer #4: No

Reviewer #5: Yes

3. Have the authors made all data underlying the findings in their manuscript fully available?

Reviewer #1: Yes

Reviewer #2: Yes

Reviewer #3: Yes

Reviewer #4: Yes

Reviewer #5: No

4. Is the manuscript presented in an intelligible fashion and written in standard English?

Reviewer #1: Yes

Reviewer #2: Yes

Reviewer #3: Yes

Reviewer #4: Yes

Reviewer #5: Yes

5. Review Comments to the Author

Reviewer #1: In this study, Jitprasertworng et al. studied the anti-inflammatory activity of soluble CHOS on VitD3-induced human THP-1 monocytes. The manuscript is well written. I have some minor questions on this project.

My Questions:

1. Why CHOS showed different functional effects between PMA induced THP-1 cells and vitamin D3 induced THP-1 cells. It will be interesting if the author could explain the underlined mechanism on this.

2. What is the “untreated” represent in figure 3?

3. The AMPK activation data is important to the mechanism study of current project. Please show it (Page 19, line 456).

Reviewer #2: Montarop et al. observed that CHOS reduced the production of multiple pro-inflammatory cytokines associated with LPS-stimulated inflammation. The statistical analysis is well written. However, there are some minor mistakes in the text, such as microphage-like cells. Thus, the manuscript should be proof read by a native speaker.

Reviewer #3: The manuscript by Jitprasertworng et al explores the anti-inflammatory effect of soluble chito-oligosaccharides (CHOS) on vitamin D3-induced human THP-1 monocytes. The authors have found out that, in the vitamin D3-induced human THP-1 monocytes, the addition of CHOS reduces the production of multiple pro-inflammatory cytokines linked with lipopolysaccharide (LPS)-stimulated inflammation in a dose-dependent manner and without affecting cell viability while observing very marginal effects of CHOS on phorbol 12-myristate 13-acetate (PMA)-differentiated THP-1 cells. My comments for the manuscript are the following:

1. The main shortcoming of the manuscript is the use of only one cell model in the whole study. The authors should perform all in vitro experiments in at least one more experimental cell model in order to draw the conclusions.

2. The manuscript lacks in vivo experiments which are crucial for such kind of studies.

3. The mechanistic part (signaling etc.) that how CHOS exerts anti-inflammatory activity on vitamin D3-induced human THP-1 monocytes is also lacking in the study.

4. The authors should check the mRNA expression of IL-1b, IL-6, and TNF-a in the LPS-stimulated, vitamin D3-differentiated THP-1 cells in the presence of various concentrations of CHOS.

5. The manuscript is not written very well, and requires an attention to improve grammar and the readability.

Reviewer #4: 1- The statistical analysis part refers to mean ± SD, ANOVA (and Post Hoc tests). However, the results of such are all presented in Fig (3-7) with no values can be accurately readable. Please, with or without the figures provide tables with (mean ± SD , sample size , exact P-values)and indicating which group is significantly different from each other's (using letters notation) Not (#) which is very stuffy.

2- Some of your results are in (%) e.g. (Fig5), (Fig7c) for which parametric measures and tests is not the proper ones, please convert to median and (25%, 75%) and use Kurskal Wallice (Instead of ANOVA) with its proper Post Hoc Test.

Reviewer #5: In the current manuscript, the authors have evaluated the anti-inflammatory activity of Chito-oligosaccharides (CHOS) using human monocyte THP-1 cells upon their differentiation to mature monocytes by Vitamin D3. The authors have provided the evidences that CHOS reduce the levels of pro-inflammatory cytokines (IL-1b, IL-6 and TNF-a) induced by Vit D3 and LPS in culture supernatants. Although some of these data are interesting, but I feel that the experiments could have been planned better and additional experiments should be performed to support their conclusions. I have following specific comments.

1. The mechanism of anti-inflammatory effect of CHOS has not been investigated enough.

2. Fig. 3. Why only and specifically IL-1b was assessed? It would be nice to also include IL-6 and TNF-a here.

3. The manuscript has many inconsistencies in writing and contains several typographical errors. For example, Line 133; I assume it should be Passage 4 (not 41)? Line 158 and 163; section 2.7 and 2.9 are mentioned, but the sections are not numbered.

4. Two different ranges for high MW CHOS have been provided in Line 72 and Line 404. Range for high and low MW CHOS should be clearly and uniformly mentioned.

5. Line 408, better to convert Da into kDa to keep uniformity.

6. Line 442-444, “CD14 is an essential co-receptor for the recognition of bacterial LPS, binding directly to LPS and presenting LPS molecules to the TLR4-MD-2 signalling complex, which leads to cellular activation”. This information should be provided on Pag 14 where you are first time introducing CD14. Please check and fix such inconsistencies at other places also.

6. PLOS authors have the option to publish the peer review history of their article (what does this mean?). If published, this will include your full peer review and any attached files.

Reviewer #1: No

Reviewer #2: No

Reviewer #3: No

Reviewer #4: No

Reviewer #5: **Yes: **Sajid Khan

---

## [Author Response · Author response to Decision Letter 0]

8 Nov 2020

Point-by-point Response to Reviewers' Comments

PONE-D-20-27129

Anti-inflammatory activity of soluble chito-oligosaccharides (CHOS) on VitD3-induced human THP-1 monocytes

PLOS ONE

Additional Editor Comments:

1- What are the final conclusion and take home message? Add this at the end of the abstract

Response: Done. The final conclusion and take-home message have been added at the end of the abstract of the revised manuscript as follows [line 78-81, 40-43; clean version];

"Altogether, in addition to showing promising biological effects of well-characterized low molecular weight soluble CHOS in a human system, the present study also points at Vitamin D3-stimulated THP-1 cells as a favorable system for assessing the anti-inflammatory activity of bioactive compounds."

2- What are the study strengths and limitations? Add this in a separate section ahead of conclusion. Start with the strength of the study followed by limitation.

Response We are grateful for this suggestion and apologize for not properly addressing these points in the original manuscript. We have rewritten and expanded the “Conclusion” section to better address the strengths and limitations of this study and also to address point 4, below, about clinical relevance and future perspectives. The rewritten “Concluding Remarks” section reads as follows (line 1078-1141, 535-562; clean version):

Concluding Remarks

 "The strength of the present study lies in the utilization of properly characterized CHOS that were prepared using (green) enzyme technology and that, in contrast to the majority of previous studies, were well characterized. The CHOS mixtures in this study have low MW, facilitating incorporation into various nutraceutical or cosmeceutical preparations. In addition, we used human cells, as opposed to commonly used murine systems, for characterizing the CHOS, leading to the conclusion that soluble low MW CHOS only act on monocytes. The selective functionality of the CHOS confirms previous observations that different structures of CHOS can have different biological activities. While our study shed light on the use of THP-1 cells for testing bioactivity, one limitation of the present study is that the biological effects of CHOS were only assessed in vitro.

In summary, we have shown that enzymatic hydrolysis of highly de-acetylated chitosan (FA 0.15) by a chitosanase from Bacillus subtilis (BsCsn46A) yielded soluble, small CHOS products with potent anti-inflammatory activity when tested on VitD3-induced THP-1 cells. A similar effect of CHOS was not observed when using PMA-induced THP-1 cells. The results underscore that the choice of the differentiation method is important when studying immunomodulatory activities of CHOS with human THP-1 monocytes and that different CHOS structures can have diverse biological activities. The strong anti-inflammatory activity of the CHOS studied here, comparable to Dexamethasone, implies clinical relevance as a novel non-steroidal anti-inflammatory substance. Importantly, the results also indicated that, while CHOS have promising medically relevant properties, the dosage of CHOS in potential pharmaceutical, nutraceutical or cosmeceutical products needs to be carefully optimized to obtain the highest benefits and prevent adverse reactions. Since chitin/chitosan and CHOS are biocompatible with low toxicity, clinical trials of soluble CHOS products for prevention and treatment of pathogenic inflammatory conditions seems an attractive next step."

3- Provide Figs in color 

Response: Done.

4- What are the clinical relevance and future perspective? Add this to the conclusion section. The clinical relevance cannot be summarized in "more prospective studies are needed...", that has no clinical relevance. Please point at how your findings impact the care of patients. 

Response: Done. We have added clinical relevance and future perspective in the expanded "Concluding remarks" section of the revised manuscript as mentioned above.

5- Make sure that Tables/Figs/references were cited in order without missing any 

Response: Done. We have checked and confirmed this.

6- Make sure that Table legends and Fig. captions are provided, without missing any

Response: Done. We have checked and confirmed this.

Journal Requirements:

Response. Done. We have checked and confirmed this.

Response. Done. We have added the data to the revised manuscript. This point has also been raised by one of the reviewers as well. This data has now become Fig. 8 of the revised manuscript as follows;

Line 1063, 521; clean version):

"Fig 8. Effects of MY-COS on AMPK activation"

5. Review Comments to the Author

Reviewer #1: In this study, Jitprasertworng et al. studied the anti-inflammatory activity of soluble CHOS on VitD3-induced human THP-1 monocytes. The manuscript is well written. I have some minor questions on this project.

My Questions:

1. Why CHOS showed different functional effects between PMA induced THP-1 cells and vitamin D3 induced THP-1 cells. It will be interesting if the author could explain the underlined mechanism on this.

Response. Done. The underlined mechanism has been written; it can be found in the revised manuscript as follows (line 711-721; 375-385; clean version);

"In summary, Fig. 4 shows that the different responses to LPS and CHOS in VitD3- and PMA-induced cells may be due to quite pronounced differences between the two differentiated cell types. It is known that stimulation of THP-1 cells with potent inducer agents such as PMA may cause phenotypic and functional differences and growth cycle arrest [21]. Accordingly, in our experiments, we found that the number of PI-positive cells increased after PMA- induced differentiation, suggesting that PMA may be cytotoxic, while there were almost zero PI-positive cells among vitamin D3-differentiated cells. Treatment of THP-1 cells with VitD3 is known to induce the cells to differentiate along the myeloid lineage to mature monocyte-like cells, resembling primary human monocytes, which are characterized by expressing a high level of CD14 and are capable of releasing a myriad of cytokines [22, 23]." 

2. What is the “untreated” represent in figure 3?

Response. Done. "untreated" is no “pre-treatment with CHOS” or 0 µg/ml of CHOS was added. We have re-labelled Fig.3 and revised the figure legend to make it clearer in the revised manuscript, accordingly. (line 643, 367; clean version).

3. The AMPK activation data is important to the mechanism study of current project. Please show it (Page 19, line 456).

Response. Done. We have added this information as Fig. 8 of the revised manuscript as mentioned above.

Reviewer #2: Montarop et al. observed that CHOS reduced the production of multiple pro-inflammatory cytokines associated with LPS-stimulated inflammation. The statistical analysis is well written. However, there are some minor mistakes in the text, such as microphage-like cells. Thus, the manuscript should be proof read by a native speaker.

Response. Agree. Thank you very much for your comment. Indeed, we have corrected the mistake in the abstract. We have double check our manuscript and corrected all typographic errors. Although Prof. Vincent Eijsink, the co-author is not a native English speaker, his English is highly proficient. He has read and edited the entire manuscript.

Reviewer #3: The manuscript by Jitprasertworng et al explores the anti-inflammatory effect of soluble chito-oligosaccharides (CHOS) on vitamin D3-induced human THP-1 monocytes. The authors have found out that, in the vitamin D3-induced human THP-1 monocytes, the addition of CHOS reduces the production of multiple pro-inflammatory cytokines linked with lipopolysaccharide (LPS)-stimulated inflammation in a dose-dependent manner and without affecting cell viability while observing very marginal effects of CHOS on phorbol 12-myristate 13-acetate (PMA)-differentiated THP-1 cells. My comments for the manuscript are the following:

1. The main shortcoming of the manuscript is the use of only one cell model in the whole study. The authors should perform all in vitro experiments in at least one more experimental cell model in order to draw the conclusions.

Response. Another human cell model in this study is human intestinal epithelial (T84) cells. The data on the study of the effects of CHOS on this experimental cell model has been added as Fig. 8 of the revised manuscript. Moreover, we would also like to point out that several analysis on the effects of CHOS on the inflammatory responses of THP-1 cell model were done in addition to the measurement of IL-1�, these are TNF-� and IL-6, CD14, as well as MTT assay to demonstrate that CHOS has no cytotoxic effect. 

2. The manuscript lacks in vivo experiments which are crucial for such kind of studies.

Response. Rebuttal. An in vivo experiment is beyond the scope of study, which focus on the production and analysis of well-characterization of CHOS and developing effective anti-inflammatory in vitro human cellular model. 

3. The mechanistic part (signaling etc.) that how CHOS exerts anti-inflammatory activity on vitamin D3-induced human THP-1 monocytes is also lacking in the study.

Response. Rebuttal. Elucidating the mechanism of action (signaling pathway) of anti-inflammatory activity of soluble CHOS is beyond the scope of this study and will be done in the next step. In this study, we demonstrated for the first time that well-characterized soluble CHOS has strong anti-inflammatory activity, comparable to Dexamethasone. Since inflammatory responses are very complex biological process, various molecular cell biology approach, including omics technique, will have to be done to elucidate mechanism of action. Nevertheless, we have begun to show that it appears that the mechanism of action is different from larger CHOS, as it doesn't act via the AMP-activated protein kinase (AMPK) in human T84 cells (Fig. 8 of the revised manuscript.

This issue has been written in the last part of the manuscript as follows;

Line 1067-1075, 526-534, clean version:

 "Inflammation and related cytokine production are complex processes and there are multiple inflammatory signaling pathways that can be inhibited by different compositions of CHOS mixtures. Further investigations must be conducted to completely understand the mechanism of the anti‑inflammatory effects of the soluble CHOS used in the present study. As it stands, the current findings underscore that different types of CHOS may have different effects and emphasize the need for well-defined CHOS preparations in biological analysis. Further analysis of the anti-inflammatory effects of CHOS in specific organs in vivo should be performed to further assess the potential of using CHOS for protective or therapeutic purposes."

4. The authors should check the mRNA expression of IL-1b, IL-6, and TNF-a in the LPS-stimulated, vitamin D3-differentiated THP-1 cells in the presence of various concentrations of CHOS.

Response. Rebuttal. Thank you very much for your suggestion. It would be interesting to explore the anti-inflammatory effect of CHOS on the production of IL-1�, IL-6, and TNF-� at both transcription and protein levels. However, mRNA expression does not always represent the protein secretion and functional effect of the cytokines. Therefore, protein level was the choice of this study. Further work on transcriptomic analysis would be interesting and informative, from which analysis at the protein level will have to be done to validate the mRNA expression study.

5. The manuscript is not written very well, and requires an attention to improve grammar and the readability.

Response. Rebuttal. We respect your opinion. Indeed, we apologize that there are some typos and minor mistake, which have been corrected in the revised manuscript. We also would like to point out that this comment is opposite from other reviewers. We are quite confident about the readability of this manuscript because it is co-authored by Prof. Vincent Eijsink, who is a renowned scient in the field. We have thoroughly gone over the manuscript for several months before submitting to the journal.

Reviewer #4: 1- The statistical analysis part refers to mean ± SD, ANOVA (and Post Hoc tests). However, the results of such are all presented in Fig (3-7) with no values can be accurately readable. Please, with or without the figures provide tables with (mean ± SD , sample size , exact P-values) and indicating which group is significantly different from each other's (using letters notation) Not (#) which is very stuffy.

Response. Thank you for these comments. The authors have adjusted the scale of the bar graph for better result illustration of Fig. 6 and put in supplementary Figure (S1 Fig.). In addition, tables of data with mean ± SD, sample size, exact P-values, have been provided as the supplementary materials of the revised manuscript (S1 Table). Letter notion has been used to indicate significantly different instead of symbol # as suggested.

Line 896-897, 452-453; clean version: A sentence "see supplementary S1 Fig. for an enlargement of the Y-axis scale" was added to the revised manuscript.

Line 964-965, 489-490; clean version: A sentence "See supplementary S1 Fig. for enlarged scale of condition without LPS stimulation (-LPS)." was added to the legend to Fig. 6 of the revised manuscript.

2- Some of your results are in (%) e.g. (Fig5), (Fig7c) for which parametric measures and tests is not the proper ones, please convert to median and (25%, 75%) and use Kurskal Wallice (Instead of ANOVA) with its proper Post Hoc Test.

Response. Thank you for these suggestions. We have re-performed statistical analysis as suggested. The revised statistical analysis is described in the materials and methods section of the revised manuscript as follows; (line 423-433, 266-276; clean version). Figure 3,4,5,6,7 have been revised with new statistical analysis.

"The data was recorded as mean � standard deviation (SD) and were analyzed by one-way and two-way analysis of variance (ANOVA). Kruskal-Wallis test and Dunn's multiple comparisons test were applied for analyses of non-parametric data of Fig. 4 - 7; while parametric data were analyzed with Brown-Forsythe ANOVA test and Unpaired t with Welch's correction test for Fig. 3, after normal distribution was indicated. Statistical testing was performed using GraphPad Prism 8 software (GraphPad Software Inc., California, USA). Tables of data with mean ± SD, sample size, exact P-values, of each experiment can be found in the supplementary materials Table S1".

Reviewer #5: In the current manuscript, the authors have evaluated the anti-inflammatory activity of Chito-oligosaccharides (CHOS) using human monocyte THP-1 cells upon their differentiation to mature monocytes by Vitamin D3. The authors have provided the evidences that CHOS reduce the levels of pro-inflammatory cytokines (IL-1b, IL-6 and TNF-a) induced by Vit D3 and LPS in culture supernatants. Although some of these data are interesting, but I feel that the experiments could have been planned better and additional experiments should be performed to support their conclusions. I have following specific comments.

1. The mechanism of anti-inflammatory effect of CHOS has not been investigated enough.

Response. Rebuttal. The focus of this research article is on the production and analysis of well-characterization of CHOS and developing effective anti-inflammatory in vitro human cellular model. This is the first report that showed that soluble CHOS in this study show strong anti-inflammatory activity of the CHOS studied here, comparable to Dexamethasone. The research on the mechanism of anti-inflammatory effect or in vivo experiment is beyond the scope of study.

This issue has been written in the last part of the manuscript as follows;

Line 1067-1075, 526-534, clean version

 "Inflammation and related cytokine production are complex processes and there are multiple inflammatory signaling pathways that can be inhibited by different compositions of CHOS mixtures. Further investigations must be conducted to completely understand the mechanism of the anti‑inflammatory effects of the soluble CHOS used in the present study. As it stands, the current findings underscore that different types of CHOS may have different effects and emphasize the need for well-defined CHOS preparations in biological analysis. Further analysis of the anti-inflammatory effects of CHOS in specific organs in vivo should be performed to further assess the potential of using CHOS for protective or therapeutic purposes."

2. Fig. 3. Why only and specifically IL-1b was assessed? It would be nice to also include IL-6 and TNF-a here.

Response. Because this is the first step of this paper. The aim of the experiment shown in Fig. 3 is to test the differentiation method, i.e. comparing PMA and VitD3 as inducer of THP-1 differentiation. As a result, we chose the VitD3 differentiation in the latter experiments.

3. The manuscript has many inconsistencies in writing and contains several typographical errors. For example, Line 133; I assume it should be Passage 4 (not 41)? Line 158 and 163; section 2.7 and 2.9 are mentioned, but the sections are not numbered.

Response. We apologize for the typos and mistakes. We have carefully checked and corrected the entire revised manuscript again already. However, passage 41 is not a mistake, this is the actual passage number of the THP-1 cells that we purchased from the company as mentioned in the materials and methods section, line 288 "(passage 41 Lot-no. 300356-1714SF) were purchased from CLS Cell Lines Service GmbH, Eppelheim, Germany." We have already mentioned in the material and methods section (line 306-307, 158-159; clean version) that "All experiments were conducted from cells in passages from 4 - 14". This passage number was count after we expand the stock that we received from the company. We feel that this information (lot number, passage number) for analysis involving THP-1 cell lines is critical; therefore, we would like to inform about the biological system that we used as much as possible.

4. Two different ranges for high MW CHOS have been provided in Line 72 and Line 404. Range for high and low MW CHOS should be clearly and uniformly mentioned.

Response. The range that we indicated was the actual range from 2 different publications (reference 14 and 29) that we cited. Indeed, this is an issue in the field that it would be nice if the scientific community can come up with a standard way to define high, medium, and low MW CHOS. Since different publications described the size of COS/CHOS differently, we intentionally informed the ranges from the previous publications in our manuscript to be precise in our publication.

Line 175-178; 78-81; clean version:

In addition, CHOS with large MW (5 -14 kDa) has been shown to inhibit NF-κB transcriptional activity, and NF-κB-mediated inflammatory responses and barrier disruption via AMPK-independent mechanisms in the human colon [14].

Line 898-901, 456-459; clean version:

High molecular weight (30-130 kDa) chitosan forms a stable water-soluble chitosan–LPS complex through electrostatic interactions. Interestingly, these complexes could enhance the secretion of TNF-� from murine monocyte cell line RAW 364.7 and IL-8 production from HEK293 cells overexpressing TLR4/MD2 [29].

5. Line 408, better to convert Da into kDa to keep uniformity.

Response. Done. Thank you very much for your suggestion. We have converted Da to kDa as suggested. 

Line 903, 461; clean version: 5000 – 300000 Da has been revised to "5-300 kDa".

Line 185, 88; clean version: 400 - 1500 Da has been changed to "0.4 – 1.4 kDa"

Line 553, 322; clean version: 400 (DP2) – 1200 Da (DP6). has been changed to "0.4 (DP2) – 1.2 kDa (DP6)". 

6. Line 442-444, “CD14 is an essential co-receptor for the recognition of bacterial LPS, binding directly to LPS and presenting LPS molecules to the TLR4-MD-2 signalling complex, which leads to cellular activation”. This information should be provided on Pag 14 where you are first time introducing CD14. Please check and fix such inconsistencies at other places also.

Response. Done. Thank you for your suggestion, we have moved this up to the part Divergent responses of THP-1 cells upon induction for differentiation by PMA and VitD3 of the revised manuscript (line 659-703, 364-367; clean version). In the original manuscript, we only mentioned that CD14 is macrophage biomarker because this part deal with differentiation. We have also checked the entire manuscript regarding this issue.

---

## [Decision Letter · Decision Letter 1]

17 Nov 2020

PONE-D-20-27129R1

Anti-inflammatory activity of soluble chito-oligosaccharides (CHOS) on VitD3-induced human THP-1 monocytes

PLOS ONE

Dear Dr. Yamabhai,

Thank you for submitting your manuscript to PLOS ONE. After careful consideration, we feel that it has merit but does not fully meet PLOS ONE’s publication criteria as it currently stands. Therefore, we invite you to submit a revised version of the manuscript that addresses the points raised during the review process.

We look forward to receiving your revised manuscript.

Kind regards,

A. M. Abd El-Aty

Academic Editor

PLOS ONE

Editor's Comments: Reviewers # 3 as well as 5 are not satisfied from the response to their comments. Authors' MUST do their efforts to respond to them. Furthermore

1- Avoid using "We, our". Use impersonal phrasing throughout the text 

2- Abbreviations MUST be stated in full acronym before being abbreviated throughout the MS

3- Don't use abbreviations in keywords. limit them to 5 to 6 keywords

4- Method performance and optimization should be provided in section "Analysis of CHOS by MS"

5- The P letter for statistical analysis should be uppercase-italic face letter. Hz should be italized as well

6-Study strength and limitations should be separated from the section "Concluding remarks". This means line 536-546 should be under heading "Study strength and limitations". The rest should be under conclusion

7- If for any reason, the authors' can't do the experimental work suggested by the diligent reviewers, this at least MUST be acknowledged under study limitation

8- Proofread the text for grammar and syntax errors

Reviewers' comments:

Reviewer's Responses to Questions

**Comments to the Author**

1. If the authors have adequately addressed your comments raised in a previous round of review and you feel that this manuscript is now acceptable for publication, you may indicate that here to bypass the “Comments to the Author” section, enter your conflict of interest statement in the “Confidential to Editor” section, and submit your "Accept" recommendation.

Reviewer #1: All comments have been addressed

Reviewer #2: All comments have been addressed

Reviewer #3: (No Response)

Reviewer #4: All comments have been addressed

Reviewer #5: (No Response)

2. Is the manuscript technically sound, and do the data support the conclusions?

Reviewer #1: Yes

Reviewer #2: Yes

Reviewer #3: (No Response)

Reviewer #4: Yes

Reviewer #5: Yes

3. Has the statistical analysis been performed appropriately and rigorously? 

Reviewer #1: Yes

Reviewer #2: Yes

Reviewer #3: (No Response)

Reviewer #4: Yes

Reviewer #5: Yes

4. Have the authors made all data underlying the findings in their manuscript fully available?

Reviewer #1: Yes

Reviewer #2: No

Reviewer #3: (No Response)

Reviewer #4: Yes

Reviewer #5: Yes

5. Is the manuscript presented in an intelligible fashion and written in standard English?

Reviewer #1: Yes

Reviewer #2: Yes

Reviewer #3: (No Response)

Reviewer #4: Yes

Reviewer #5: Yes

6. Review Comments to the Author

Reviewer #1: In this study, Jitprasertworng et al. studied the anti-inflammatory activity of soluble CHOS on VitD3-induced human THP-1 monocytes. The manuscript is well written. The authors have addressed all my concerns. I do not have further questions on it.

Reviewer #2: All my questions has been addressed. The manuscript can be accepted in current version.

Reviewer #3: (No Response)

Reviewer #4: (No Response)

Reviewer #5: Although the authors have not performed any additional experiments suggested to elucidate the mechanism of anti-inflammatory effect of CHOS, they have adequately addressed my other concerns. I do not have any further comments.

7. PLOS authors have the option to publish the peer review history of their article (what does this mean?). If published, this will include your full peer review and any attached files.

Reviewer #1: No

Reviewer #2: No

Reviewer #3: No

Reviewer #4: No

Reviewer #5: No

---

## [Author Response · Author response to Decision Letter 1]

3 Dec 2020

03 December 2020

Dear Editor

Ref. PONE-D-20-27129R2

Anti-inflammatory activity of soluble chito-oligosaccharides (CHOS) on VitD3-induced human THP-1 monocytes

Thank you very much for checking our manuscript.

I am sending you the revised version of the manuscript regarding the financial disclosure issue following your advices. I have removed all funding-related text from the acknowledgement section. And I would like to update the Funding Statement as shown below (the bold text is the new text that I would like to add).

"This research was supported by the Suranaree University of Technology (SUT) [grant no. RU12/2562], the National Research Council of Thailand (NRCT) and the Office of the Higher Education Commission (OHEC) under the National Research University (NRU) project [grant no. SUT3-304-55-36-18 and SUT3-304-62-36-18]. MK and KR were supported by SUT Full-time Doctoral Research [grants no. 61/13/2561 and 61/28/2563] and Thailand Science Research and Innovation (TSRI). PP was supported by a SUT-PhD scholarship."

In addition, I have submitted the raw western blot (WB) data, which was taken from ChemiDoc machine and labeled using Adobe Illustrator without adjusting or cropping. These results were prepared by the scientist from the laboratory of one of the co-authors (Chatchai Muanprasat).

I hope the manuscript ready for publication, otherwise please kindly let us know.

Thank you very much for kind your considerations.

---

## [Editor Report · Decision Letter 2]

4 Dec 2020

PONE-D-20-27129R2

Anti-inflammatory activity of soluble chito-oligosaccharides (CHOS) on VitD3-induced human THP-1 monocytes

PLOS ONE

Dear Dr. Yamabhai,

Thank you for submitting your manuscript to PLOS ONE. After careful consideration, we feel that it has merit but does not fully meet PLOS ONE’s publication criteria as it currently stands. Therefore, we invite you to submit a revised version of the manuscript that addresses the points raised during the review process.

ACADEMIC EDITOR: Author's should respond to the comments raised by reviewer # 3 as stated below

We look forward to receiving your revised manuscript.

Kind regards,

A. M. Abd El-Aty

Academic Editor

PLOS ONE

Additional Editor Comments:

Author's MUST respond to the comments raised by reviewer # 3 as stated below

The authors have not satisfactorily addressed the comments. In the previous version of the manuscript, they had used only one cell line model for the study. Therefore, they were asked to add at-least one more experimental cell model in order to draw the concrete conclusions. In the revised manuscript, there are no crucial data added using another cell line.  In T84 cells, the only data I could find was the effect of short CHOS on AMPK activation. Moreover, they have added some of the contradictory statements in the revised version of the manuscript. For example, in context of mechanism (signaling), they have stated that (Line-517-519) “The results showed that the short CHOS used in the present study did not activate AMPK in T84 cells compared to a known AMPK activator, A-769662, that was used as a positive control”. But, when we carefully look at the Figure 8, we do see a clear activation of AMPK when normalized with total-AMPK in all the replicates.

---

## [Author Response · Author response to Decision Letter 2]

15 Jan 2021

As pointed out before, we respectfully argue that it is not reasonable to ask for a complete set of data for another cell line model. This paper has a strong, and relatively unique, focus on human macrophages. We included some results with T84 cells in the original version of the manuscript as "data not shown". In response to the comments on the original manuscript, in the revised manuscript, we showed the Western blot data and the quantification as presented in Fig. 8. We acknowledge that this is rather limited data, but we feel that it is not reasonable to ask for more. The other 4 reviewers were satisfied with our responses and revisions.

 We do, however, understand the confusion that Fig. 8 has created, as pointed out by the reviewer. We are grateful to the reviewer for bringing this up. We agree with reviewer #3 that by looking at the figure, it would seem that CHOS did activate AMPK; however, by quantification from three replicates, we found that the differences (i.e., the potential activation) were statistically insignificant (albeit close to significant). This situation is not satisfactory, and the reviewer comment thus has merit. Therefore, we have repeated the experiments two more times (reaching a total of n=5). Indeed, the combined results of the five experiments indicated that CHOS does activate AMPK. The revised parts of the manuscript read (Significantly changed parts with yellow shading):

Line 524-583 (499-543, clean version)

"Longer CHOS with MW of > 5 kDa have been reported to activate AMP-activated protein kinase (AMPK) [14], and this effect could, at least in part, explain the previously observed anti-inflammatory effect of such CHOS in human intestinal epithelial (T84) cells [2, 14]. Thus, this biological activity was investigated using the soluble CHOS employed in the present study (100 �g/mL; 24 h incubation). The results show that, these short CHOS activate AMPK in T84 cells, similar to the effect of a known AMPK activator, A-769662, that was used as a positive control (Fig. 8). Indeed, it has been reported that AMPK activation suppresses proinflammatory responses in mouse and human macrophages [32]. Therefore, one of the possible mechanisms behind the observed anti-inflammatory action of the soluble CHOS studied here is via the activation of AMPK.

Fig. 8. Effects of CHOS on AMPK activation. Western blot (WB) analysis of p-AMPK, AMPKα, and β-actin after treating T84 cells with CHOS at 100 μg/mL for 24 h. Data are analyzed as the ratio of p-AMPK/AMPKα and expressed as fold change in the ratio compared to the ratio for the control group (means ± SEM, n = 5). The compound A-769662 is a known activator of AMPK. c = P < 0.05 (compared with control). 

 Inflammation and related cytokine production are complex processes and there are multiple inflammatory signaling pathways that can be inhibited by different compositions of CHOS mixtures. Further investigations must be conducted to completely understand the mechanism of the anti‑inflammatory effects of the soluble CHOS used in the present study. Further analysis of the anti-inflammatory effects of CHOS in specific organs in vivo should be performed to further assess the potential of using CHOS for protective or therapeutic purposes."

Please note that one reference has been added, number 32: Sag D, Carling D, Stout RD, Suttles J. Adenosine 5'-monophosphate-activated protein kinase promotes macrophage polarization to an anti-inflammatory functional phenotype. J Immunol. 2008;181(12):8633-41. doi: 10.4049/jimmunol.181.12.8633. PubMed PMID: 19050283.

The revised Fig. 8 has been uploaded as part of this submission, while the additional uncropped figures have been added to the Supplementary section of the submitted-revised manuscripts. 

We would like to point out that these changes do not alter the main message of this paper, but rather provide one possible mechanism behind the anti-inflammatory activity of soluble CHOS.

---

## [Decision Letter · Decision Letter 3]

18 Jan 2021

Anti-inflammatory activity of soluble chito-oligosaccharides (CHOS) on VitD3-induced human THP-1 monocytes

PONE-D-20-27129R3

Dear Dr. Yamabhai,

We’re pleased to inform you that your manuscript has been judged scientifically suitable for publication and will be formally accepted for publication once it meets all outstanding technical requirements.

Kind regards,

A. M. Abd El-Aty

Academic Editor

PLOS ONE

Additional Editor Comments (optional):

Reviewers' comments:

Reviewer's Responses to Questions

**Comments to the Author**

1. If the authors have adequately addressed your comments raised in a previous round of review and you feel that this manuscript is now acceptable for publication, you may indicate that here to bypass the “Comments to the Author” section, enter your conflict of interest statement in the “Confidential to Editor” section, and submit your "Accept" recommendation.

Reviewer #3: All comments have been addressed

2. Is the manuscript technically sound, and do the data support the conclusions?

Reviewer #3: Yes

3. Has the statistical analysis been performed appropriately and rigorously? 

Reviewer #3: Yes

4. Have the authors made all data underlying the findings in their manuscript fully available?

Reviewer #3: Yes

5. Is the manuscript presented in an intelligible fashion and written in standard English?

Reviewer #3: Yes

6. Review Comments to the Author

Reviewer #3: In the revised version of the manuscript (PONE-D-20-27129R3) by Jitprasertwong/Khamphio/Petsrichuang et al, the authors have now considerably addressed my comments to a certain extent. The manuscript may be considered for publication.

7. PLOS authors have the option to publish the peer review history of their article (what does this mean?). If published, this will include your full peer review and any attached files.

Reviewer #3: No

---

## [Editor Report · Acceptance letter]

22 Jan 2021

PONE-D-20-27129R3 

Anti-inflammatory activity of soluble chito-oligosaccharides (CHOS) on VitD3-induced human THP-1 monocytes 

Dear Dr. Yamabhai:

I'm pleased to inform you that your manuscript has been deemed suitable for publication in PLOS ONE. Congratulations! Your manuscript is now with our production department. 

Kind regards, 

on behalf of

Prof. A. M. Abd El-Aty 

Academic Editor

PLOS ONE